# Opportunistic Multi-Technology Cooperative Scheme and UAV Relaying for Network Disaster Recovery

**Farouk Mezghani** *,† and **Nathalie Mitton** †

Inria Lille—Nord Europe, 40 Avenue Halley, 59650 Villeneuve d'Ascq, France; nathalie.mitton@inria.fr
* Correspondence: farouk.mezghani@inria.fr; Tel.: +33-3-59-57-79-43
† These authors contributed equally to this work.

**Abstract:** Disaster scenarios are particularly catastrophic in urban environments, which are very densely populated in many cases. Disasters not only endanger the life of people, but also affect the existing communication infrastructures. In fact, such an infrastructure could be completely destroyed or damaged; even when it continues working, it suffers from high access demand to its limited resources within a short period of time. This work evaluates the performances of smartphones and leverages the ubiquitous presence of mobile devices in urban scenarios to assist search and rescue activities following a disaster. Specifically, it proposes a collaborative protocol that opportunistically organizes mobile devices in multiple tiers by targeting a fair energy consumption in the whole network. Moreover, it introduces a data collection scheme that employs drones to scan the disaster area and to visit mobile devices and collect their data in a short time. Simulation results in realistic settings show that the proposed solution balances the energy consumption in the network by means of efficient drone routes and smart self-organization, thereby effectively assisting search and rescue operations.

**Keywords:** disaster recovery; mobile devices; smartphone performances; multi-tier cooperative communication; drone-based data relaying; performance evaluation; energy-efficient communication

## 1. Introduction

Natural disasters such as earthquakes, tsunamis, volcanic eruptions, and flooding cause substantial damage in terms of both human lives and infrastructure costs, especially in densely populated urban environments. The first 72 h after a disaster are particularly critical—in fact, they are referred to as the *golden relief time*—and they are exactly when exhaustive research and rescue activities are taking place [11,36]. Unfortunately, such activities are usually carried out by volunteers with limited access to specialized tools and an adequate supporting infrastructure. In particular, communication networks (e.g., cellular base stations) could be completely destroyed or damaged; even when they continue working, they suffer from high access demand to their (limited) resources within a short period of time. This, in turn, exposes both people and rescue teams to the denial of communication services.

Indeed, disaster scenarios pose crucial questions regarding the most efficient way to establish communication in terms of time, energy, cost, and practicality. In particular, survivors must be able to send out emergency requests (including location data) and heartbeat-like messages but also to receive some fortifying information. Given the light-weight nature of such messages, leveraging spontaneously smart devices owned by the survivors is one possibility. Moreover, opportunistic systems relying on smart devices (nodes) can fully take advantage of their intrinsic heterogeneous-

and ubiquitous-nature. On the other hand, the rescue teams must be able to timely make use of such information and lead rescue operations efficiently.

In this respect, UAV (Unmanned Aerial Vehicles) such as drones, have been employed to provide an on-demand communication infrastructure for disaster scenarios [1–3]. Drones are particularly suitable for such situations as they can quickly and easily cover affected areas. However, using drones as a communication infrastructure for disaster recovery raises two key challenges. First, they should be able to reliably discover survivors as soon as possible. This also means that energy-efficient communication protocols should be in place to extend the time devices owned by survivors can be used for emergency response. Second, they should cover the affected area in the shortest possible time, and many survivors as possible.

Data dissemination under missing or damaged communication infrastructure has received increasing attention in the last few years [4–7]. However, most of these solutions are based upon a single communication interface (e.g., WiFi), and assume a certain level of cooperation and organization of rescue teams in the territory, thereby complementing (instead of replacing) the existing communication services. Exploring and extending the coverage of wireless communications with UAVs has also been widely investigated, especially in situations where human lives would be endangered and people cannot be physically reached [1–3]. However, most of recent solutions [8,9] solely address placement and optimization of UAVs to offer wireless communication in specific areas, and do not consider the communication aspects related to the end devices, for instance, in terms of energy efficiency.

This work specifically addresses these issues by leveraging the ubiquitous presence of mobile devices in urban scenarios to assist search and rescue activities following a disaster. Specifically, it proposes a collaborative protocol that opportunistically organizes mobile devices in multiple tiers by targeting a fair energy consumption in the whole network. In doing so, it spontaneously relies on the multiple radio interfaces in off-the-shelf personal mobile devices for energy-efficient operations. Moreover, it introduces a complementary data collection scheme that employs drones to visit mobile devices and collect their data in a short time. The main contributions of this work are the following.

- It formally characterizes the creation of a multi-tier communication infrastructure of mobile devices with multiple radio interfaces. It then derives a heuristic for clustering nodes based on their local connectivity and available energy.
- It evaluates the performance of smartphones in terms of network interfaces (based on energy consumption and transmission range) and clock synchronization.
- It introduces a scheme for drone-based data collection that minimizes the total flying path, while still ensuring a sufficient time to collect data. In particular, it derives the locations that a hovering drone needs to reach and stop at to collect data from mobile devices based on a multi-tier network structure.

Extensive simulations in realistic settings demonstrate that the proposed solution balances the energy consumption in the network by means of efficient drone routes, thereby effectively assisting search and rescue operations.

## 2. Related works

Most of the work in wireless sensor networks with mobile elements and ad-hoc networks aim at providing wireless communication during natural disaster phenomena [7,10–12]. Moreover, the ubiquitous nature of smart devices such as smartphones is largely exploited by many works, whose main focus consists of extending the wireless connectivity coverage in areas with missing or damaged infrastructure [5,13].

Smart devices constitute one key element in survivor-rescuer systems [11,14]—they send out location data of the survivors to rescue teams, for instance. Accordingly, the works in [1,3] show how such entities communicate with each other via aerial base stations, i.e., UAVs that fly over a disaster area with on-board femto-cells. However, such works provide no considerations on the energy-efficiency of the proposed solutions.

Other works with flying ad-hoc networks present performance tradeoffs as a function of parameters such as the UAV height and placement to maximize the coverage in a multi-UAV system [2,15,16]. By contrast, we focus on a single-UAV system which requires no synchronization, placement map over a given area, or task scheduling among UAVs. Moreover, our approach leverages the heterogeneity and ubiquitous nature of smart devices (representing survivors) to build a cooperative scheme underlaying the UAV. In fact, such a scheme results in a larger number of alive nodes over time, hence ensuring a wider coverage area from which nodes can disseminate their data and ask for help; at the same time, it reduces the flying time (number of stops) of an UAV over the area of interest, hence its energy consumption.

Recently, interesting works have appeared addressing the drone trajectory optimization regarding different parameters (altitude, energy consumption and harvesting, throughput, etc) [32–35]. These approaches do not couple their approach with a dynamic ground device organization but we believe they constitute a complementary approach to ours that deserves to be further investigated.

While most of the current work leverages only few of the available network technologies to build their solutions upon [7], the work in [17] exploits all such interfaces for alert diffusion during disasters. However, it only mitigates the energy expenditure of the nodes by scheduling shorter wake-up periods for nodes with low available energy levels. By contrast, this work devises a cooperative and multi-tier communication scheme that achieves energy fairness among the nodes by designating only few nodes to switch on their interfaces and relay the data of the other nodes in the network. Moreover, such nodes vary over time, hence fairly distributing the energy expenditure across all the nodes in the network.

## 3. Multi-Technology Cooperative Communication and Drone Data Relaying

### 3.1. Multi-Technology Network Architecture

This work proposes COPE, a cooperative communication scheme that leverages mobile devices (here, each device (or node) is supposed to be owned by a survivor, therefore, the two terms are used interchangeably in the rest of the article) involving multiple network technologies and characterized by various energy levels. The considered network environment consists of mobile devices equipped with multiple network technologies, such as those available in off-the-shelf smartphones (e.g., Bluetooth, WiFi, and cellular). These technologies are characterized by different transmission ranges and energy consumption characteristics [17]. Accordingly, a multiple-tier architecture is spontaneously created by opportunistically grouping devices capable of reaching each other directly (i.e., in a single hop) into clusters, as illustrated in Figure 1. The devices in each tier all use the same communication technology, and tiers are layered depending on their features. In particular, the lowest tier is the one with the most energy-efficient communication technology, but also with the shortest range. The highest tier is the one with the most energy-hungry communication technology, which also corresponds to the highest range. Intermediate tiers are made by increasing levels of energy efficiency and decreasing transmission ranges. The proposed network structure is flexible enough to include a varying number of tiers. However, the figure shows a network composed of three tiers, corresponding for instance to communication technologies existing in nowaday mobile devices such as smartphones; i.e. Bluetooth ($n_1$), WiFi ($n_2$) and cellular ($n_3$) communication technologies. This is also the most realistic option in practice, given currently available smartphones.

One node in each cluster is designated as cluster-head (CH). The CH is the node that is able to act as a bridge between different tiers: it collects data from one tier and relays them to the upper tier. The CHs in the highest tier communicate directly with a drone that hovers over them. For example, node $s_4$ in Figure 1 is a CH for the cluster that includes nodes $s_5$ and $s_6$ in the $n_1$ tier. Instead, node $s_2$ is a CH in three clusters: the one that includes nodes $s_1$ and $s_3$ in the $n_1$ tier, and the one that contains node $n_4$ in the $n_2$ tier.

In addition to the mobile devices, the network also includes drones that are sent on-demand to the area of the disaster. In particular, drones are equipped with the same technology that constitutes

the highest tiers. For instance, they can be equipped with on-board femto-cells, and provide ad-hoc cellular communication to the nodes in the highest tier [18,19]. In particular, a drone makes a tour of the network by reaching certain designated locations, where it collects data from one or more nodes, depending on the specific path planning algorithm employed (refer to Section 5 for more details). Going back to the previous example, node $s_2$ is the only one able to communicate with the drone in the $n_3$ tier among all nodes in the clusters it belongs to.

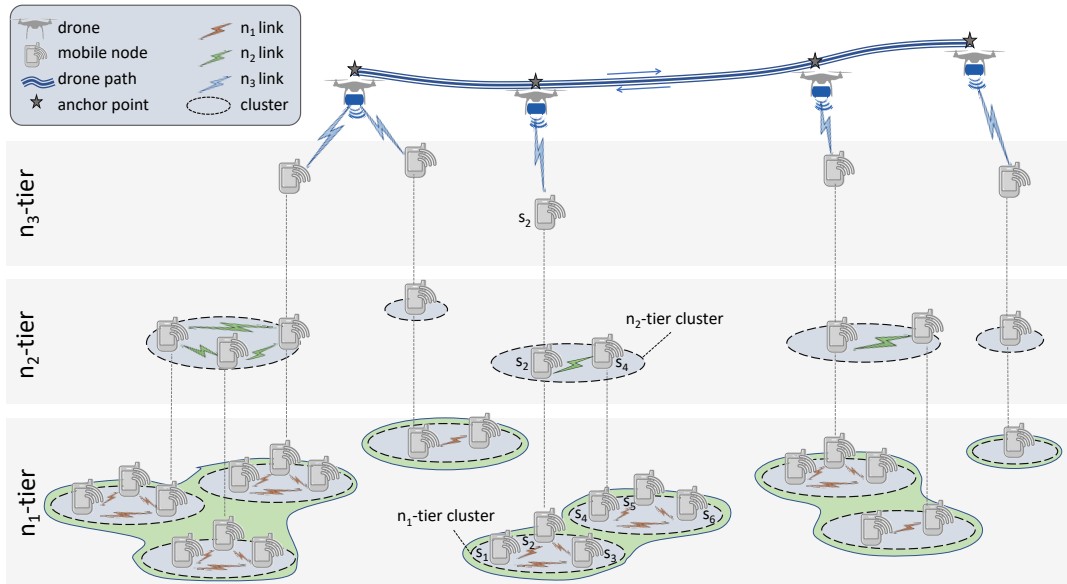

**Figure 1.** Multi-tier network architecture: (example of three communication technologies).

### 3.2. System Model

The system includes the set $\mathcal{S} = \{s_1, s_2, \ldots, s_M\}$ of $M = |\mathcal{S}|$ nodes representing survivors at their locations. The node density is relatively sparse, i.e., nodes might not all be connected; the cluster formation among multiple nodes is not guaranteed and clusters may consist of a single node. Moreover, each node is characterized by an initial available (battery) energy level $e_{s_m}$. The system also comprises the set $\mathcal{N} = \{n_u, 1 \leq u \leq U\}$ of communication interfaces, with $U$ the number of available interfaces. Furthermore, the communication interfaces are ordered based on their energy consumption $c$ as $c_{n_1} < c_{n_l} < c_{n_U}$, and transmission range $r$ as $r_{n_1} < r_{n_l} < r_{n_U}$. This assumption will be investigated in next section 4 which evaluates the performances of smartphones in terms of energy consumption and transmission range. Mobile devices are carried by survivors who move slowly if at all. This assumption is justified by the high chance that the survivors are unable to walk fast or run, due to possible injuries and the many obstacles that the natural disaster causes.

The drone collects data from survivors and make them available to search and rescue teams [18,20,21]. The drone operates in two phases: it first identifies the location of the nodes in the area affected by the disaster and plans a path that visits all the discovered nodes; it then flies around the area according to such a path. The drone moves with a fixed speed between the intended locations, where it stops for a certain amount of time. This can easily be accomplished, for instance, by using a rotary-winged drone. More details about drone data relaying are provided in Section 5.

### 3.3. Multi-Technology Communication Algorithm

The main intuition behind the proposed heuristic is that cluster formation can opportunistically leverage the local connectivity of nodes at the different tiers; CHs can then be selected to uniformly spread energy consumption between nodes, both over clusters and tiers. CHs need to transmit data over network interfaces $n_u$ $u \leq U$. Hence, they end up consuming energy faster than the other nodes. To maximize clusters lifetime, nodes within the same cluster take turns becoming cluster head for

a time interval $\delta t$ according to their energy level (i.e., the node having the highest energy level will be the CH). The heuristic is described in Algorithm 1.

---

**Algorithm 1:** Dynamic CH selection run at each node $m$.

---

1　**Input:** nodes location and $\delta t$ (time-slot)
2　**Output:** switch ON/OFF interface $n_i$
3　**foreach** $\delta t$ **do**
4　　$i = 1$;
5　　activate interface $n_1$;
6　　ACTIVATED = TRUE;
7　　**while** *(ACTIVATED==TRUE) and (i < U)* **do**
8　　　discovers $n_i$-neighbors and power budget;
9　　　select $n_i$ tier CH with higher power-budget (potentially itself);
10　　**if** $CH_i = m$ **then**
11　　　*#The node is the CH for $n_i$-tier*activate $n_{i+1}$ interface;
12　　　$i$++;
13　　**else**
14　　　ACTIVATED == FALSE
15　　**if** $i = U$ **then**
16　　　exchange data with UAV

---

**Complexity.** A node only has to identify cliques within its neighborhood (in $O(n.log(n))$ if $n$ is the number of neighbors) and sort the nodes of the clique according to their Id, so in $O(m.log(m))$ where $m$ is the number of nodes in the clique. Determining close bounds for $n$ and $m$ is a very interesting study that we may complete finely. Nevertheless, an upper bound for $m$ is $n$ ($m \leq n$). So an upper bound for the complexity is $O(n.log(n))$.

## 4. Mobile Devices Performances: Smartphone Use Case

Testing experiments have been carried out to measure the performance of smartphones in terms of network interfaces (based on energy consumption and transmission range) and clock synchronization. We believe that such experimental results can support technological choices for rescue operations but also for many other applications relying on smartphone performances.

We have conducted different experiments featuring six smartphones Wiko Tommy 2 and exploiting two network technologies Bluetooth and Wi-Fi. The main specifications of used smartphones are shown in Table 1. Testing scenarios target to measure and evaluate the performances of the smartphones using COPE application and considering the energy consumption, clock synchronization and transmission range metrics.

**Table 1.** Smartphones main specifications.

| Smartphone Model | Wiko Tommy 2 |
| --- | --- |
| OS | Android 7.1 (Nougat) |
| Battery | Li-Po 2500 mAh 9.5 Wh |
| Bluetooth | 4.1, A2DP, LE |
| WiFi | WiFi Direct |

*4.1. Energy Consumption and Transmission Range*

COPE proposes a multi-technology cooperative communication for a fair energy consumption in the whole network. Experiments were conducted to measure the energy consumption based on a non-cooperative and cooperative network topologies as follows:

($i$)　a non-cooperative communication scheme considering only one node that operates individually; i.e., nodes switch on their network interfaces (Bluetooth and WiFi direct) for communication

(*ii*) a cooperative communication scheme (i.e., COPE) considering two and three nodes respectively; i.e., nodes form groups based on the Bluetooth, then, periodically only one node turn on its WiFi interface at the same time to communicate

For measurements, only COPE application was running on smartphones with the screen turned off.

Figure 2 illustrates the energy consumption over five minutes from the WiFi and Bluetooth perspective considering the three network topologies. Results show that when a node operates individually, it consumes more energy than in the other topologies where nodes are cooperating. Moreover, as we increase the number of nodes within the group, the energy consumption is reduced since nodes will be in a sleep mode for a longer period from the WiFi perspective. Therefore, a cooperative scheme can help to reduce the energy consumption and thus to keep mobile devices alive for a longer time. We would like to emphasize that these experiments validate that, in addition, Bluetooth consumes less energy than WiFi in the context of applications requiring an exchange of small data (e.g., short text message). In the context of COPE that only needs to send short SOS messages, this confirms our assumptions.

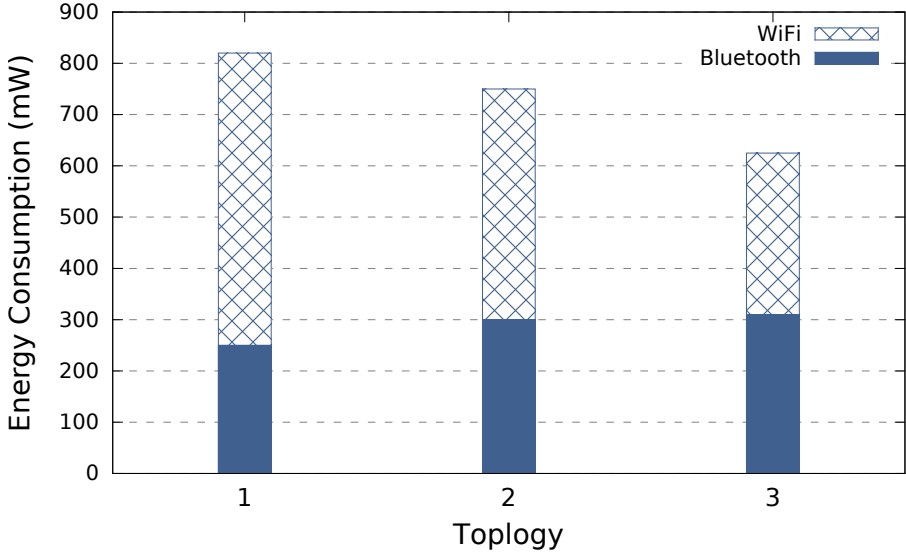

**Figure 2.** Energy consumption.

We have carried out experiments to test the transmission ranges of Bluetooth and WiFi-Direct. Testing the quality of the link and the transmission speed are not in the scope of this work (See [22] for a related study). Indeed, we consider the necessity of exchanging short text messages that can be useful for alerting or asking for assistance. Thus, in our testing scenarios, we simply try to exchange some short messages of few bytes between smartphones and we keep increasing the distance between mobile nodes until the link interruption. We have carried out various testing scenarios considering: windy and humid weathers; calm and dry weathers; outdoor line of sight (see yellow lines on Figure 3); indoor with obstacles (1 to 2 walls); between two buildings with a distance of about 80 m (see red line on Figure 3).

Table 2 presents the transmission range of the Bluetooth and Wi-Fi Direct for outdoor and indoor scenarios. These results show the importance of the new version of Bluetooth offering an important transmission range comparing to the previous versions. Several research works consider the transmission range of Bluetooth of around 10 m while the new version of Bluetooth (Bluetooth Low Energy BLE) offers more important transmission ranges compared to what is expected even in theory. Moreover, experiments validate the assumptions considered in COPE solution that WiFi offers a higher transmission range than Bluetooth.

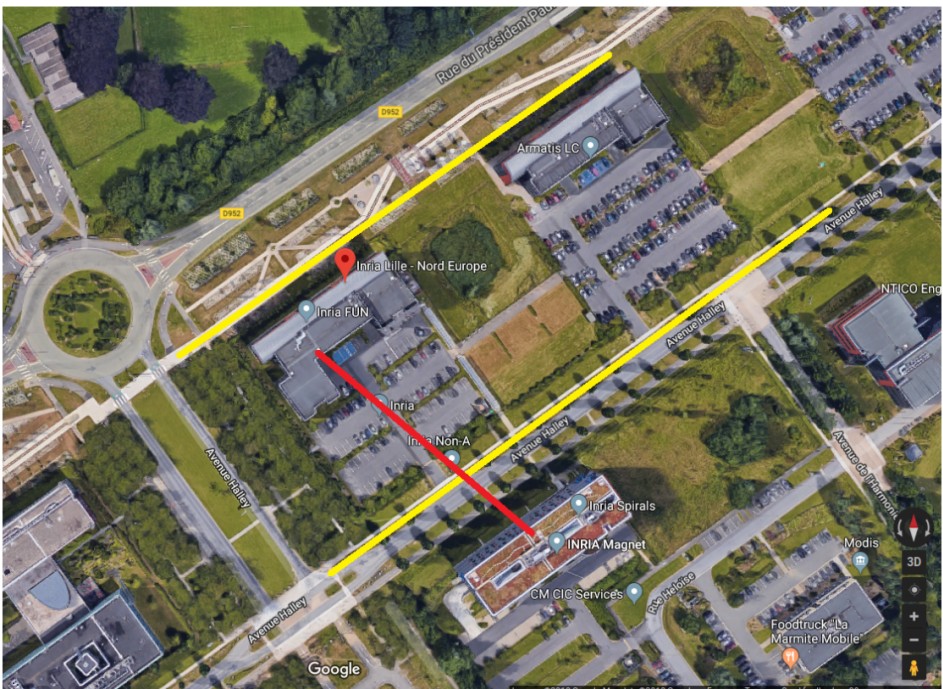

**Figure 3.** Transmission range testing area.

**Table 2.** Bluetooth and Wi-Fi Direct transmission range.

|         | Bluetooth | Wi-Fi Direct |
| ------- | --------- | ------------ |
| Indoor  | 35 m      | ≥100 m       |
| Outdoor | 50 m      | ≥100 m       |

### 4.2. Clock Drift

COPE assumes that mobile nodes are already synchronized since smartphones get the local time from the network providers with millisecond accuracy before disasters occur. In the following, we study the clock drift of the smartphones to check weather an additional synchronization is required during the post-disaster period. We have carried out a first simple experiment to check weather smartphones belonging to different network operators are synchronized. Results have shown that smartphones are a few milliseconds apart. Next, we have conducted an experiment to test the clock drift featuring the six smartphones. We initially synchronized all the mobile phones through Internet via a NTP (Network Time Protocol) time server. Afterwards, we prevent the automatic synchronization and we measure the clock drift referring to the NTP time server.

Figure 4 shows the clock drift of the different smartphones during a period of 24 h. Results show that mobile phones desynchronize by up to 0.3 s during 1 day which is not significant drift and does not impact the COPE scheme. Therefore, since the time-slot $\tau$ is at second level, COPE does not require an additional synchronization. We would like to emphasize that we have repeated the experiment considering various scenarios: smartphone display ON/OFF, smartphone in charge/not in charge and by running applications in parallel. We have obtained similar clock drift results. Smartphones synchronize their clock time with the cellular infrastructure. When disconnected from the cellular network, clock drift is not significant while difficult to predict. Indeed, results show that smartphones present different desynchronization behaviors even though we use the same mobile device model (i.e., Wiko Tommry 2).

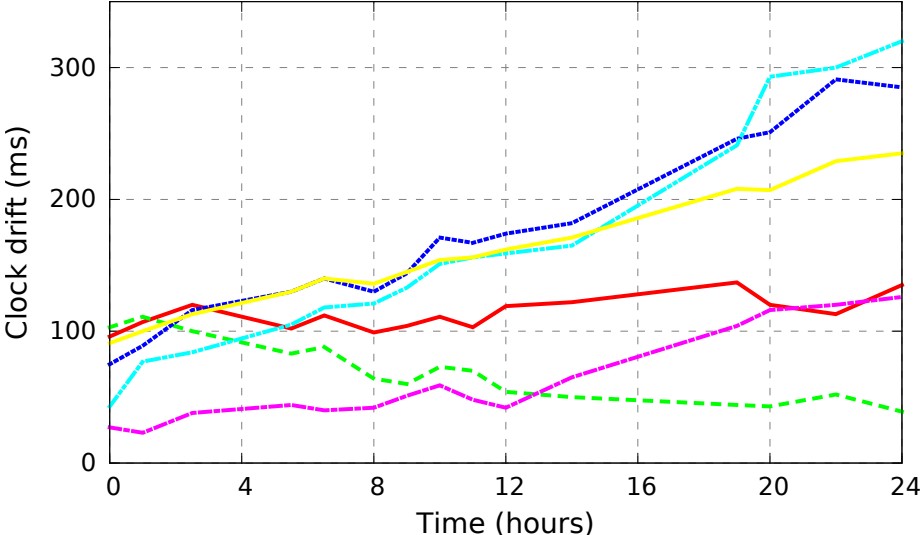

**Figure 4.** Smartphones clock drift.

## 5. UAV Data Relaying

During disaster, using an additional way of communication to relay data between rescue teams and different zones in the disaster area would ease and speed up the rescue operations. Thus, UAVs have gained increasing attention as they can move easily from one place to another and they can communicate and relay data with other devices. However, UAVs are characterized by their high cost and limited battery lifetime. Therefore, it is of great importance to efficiently use UAVs such to optimize their paths when relaying data.

Data collection as well as reporting to rescue teams leverages drones equipped with femto-cells as an on-demand communication infrastructure [8,16]. To fully cover the area affected by the disaster and effectively provide wireless communication capability, the drone must visit all the nodes which have switched on the upmost tiers interface (i.e., nodes in the $n_U$ tier). The solution proposed in this work operates in two phases.

- *Search.* A drone flies over the area affected by the disaster so as to discover nodes and store their location. The drone follows an *S*-shaped route, whose curvature guarantees that all nodes can be discovered (see Figure 5).
- *Anchor points derivation and path planning.* Once the nodes are discovered, anchor points are then derived. Anchor points can be either $n_U$ tier nodes or locations from which a drone can reach multiple $n_U$ tier nodes, if possible. That is, an anchor point can be anywhere in between the $n_U$ tier nodes it serves (Figure 6). Hence, there is no need for the drone to hover above each $n_3$ tier node—hovering above the (fewer) anchor points suffices to serve all $n_U$ tier nodes. Consequently, given such anchor points as an input to a path planning algorithm, the shortest path that visits all these points is then constructed. The drone then follows such a path and collects data (Figure 7).

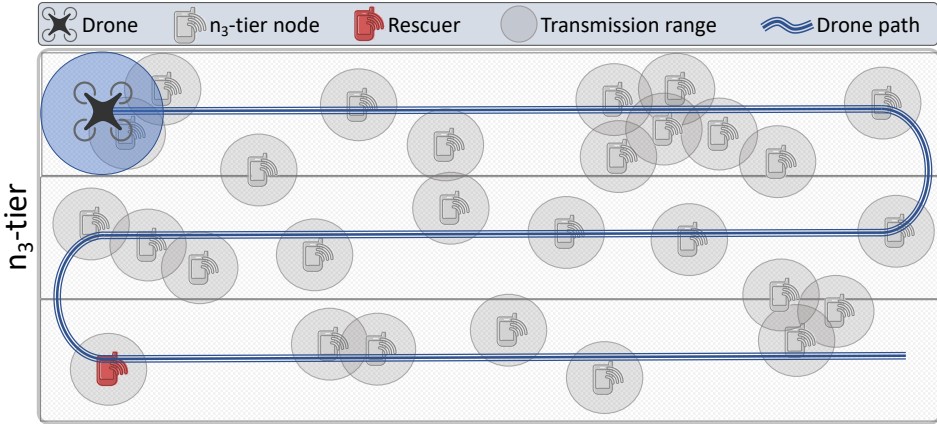

**Figure 5.** During the search phase a drone flies over the area affected by a disaster and stores the location of the discovered nodes in the highest (i.e., $n_3$) tier of the network.

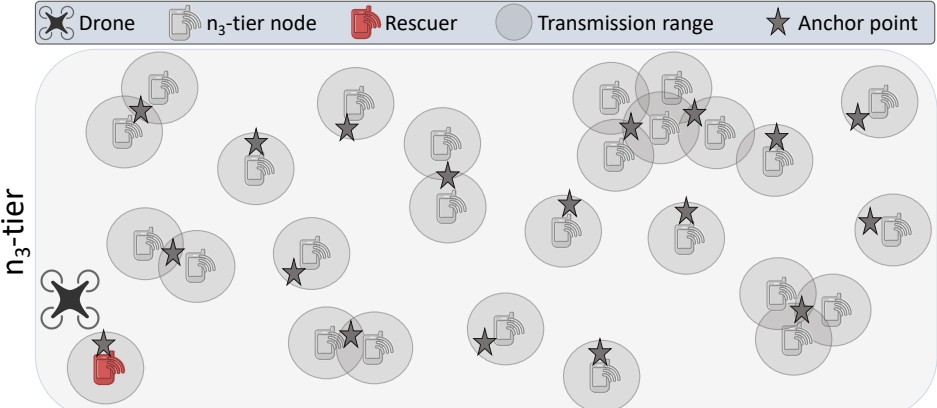

**Figure 6.** Drone anchor points shown with stars.

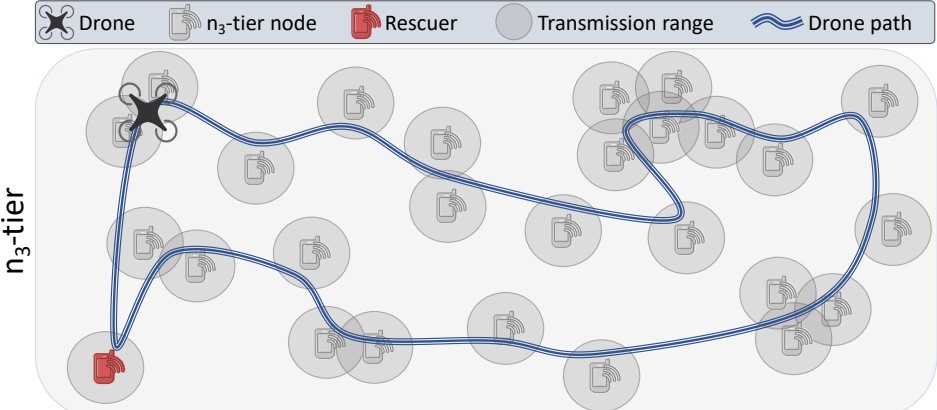

**Figure 7.** Shortest path for a drone to visit all the anchor points.

Different schemes to plan the drone's path that dictate the order of visit of the points are considered. Such schemes aim at finding the shortest path that visits all such points, while the drone must return to its initial location (to recharge, for instance). Such a problem, in fact, corresponds to the well-known NP-hard TSP problem. Other versions such as TSPN (TSP with Neighborhoods), CETSP (Close-Enough TSP), Covering Tour Problem, and Generalized TSP are extensively studied in literature [23,24].

The design of the path planning algorithms focuses especially on the energy consumption of the drone. That is, such algorithms aim at reducing the tour length of an UAV, hence the time it takes to fly

over a disaster area and collect the data from the nodes. In fact, such reduces the energy expenditure of the UAV. More specifically, leveraging the cooperative communication and data relay protocol among the nodes underlaying the UAV yields to reducing the number of points a drone should visit. In fact, identifying anchor points from which a drone can serve more than one node results in a lower number of stops for the drone. Feeding such anchor points i to the existing TSP and CETSP algorithms [23,24] comes down to construct the shortest path for a drone to follow, as shown in Figure 7.

## 6. Performance Evaluation

A performance evaluation of the proposed multi-tier data relaying scheme is conducted next, with focus on two main significant aspects: (*i*) efficiency of relaying data through the tiers in terms of energy expenditure, and (*ii*) efficiency of deploying an UAV to provide on-demand wireless communication in disaster scenarios. Specifically, experimental results are obtained through trace-driven simulations in a realistic disaster scenario. Each simulation is replicated ten times and the average values along with the related standard deviations are then reported when meaningful.

*6.1. Cooperative Multi-Tier Data Relaying*

6.1.1. Methodology and Setup

We assess the performance of the proposed multi-tier data relay as in Algorithm 1. For comparison purposes, we present the performance of other two schemes, namely baseline and static, along with our proposed solution.

- *Baseline approach.* It considers every node as a cluster, namely, each node is responsible to switch on all the necessary network interfaces to transmit its own data. Such a scheme provides no collaboration among nodes. In fact, all nodes are exposed to a maximum energy expenditure, which leads to fast battery depletion. Consequently, the chances of a node to keep in contact with search and rescue teams for long periods of time are subject to such a limitation.
- *Static approach.* The nodes collaborate among each other to relay their data through the tiers. For such a purpose, in the $n_1$ and $n_2$ tier, nodes are organized into clusters and only one *responsible* node per cluster relays data to the upper tier. Consequently, the other cluster members do not need to switch on the next communication interface, hence mitigating their energy consumption. The CHs are selected based on the initial information on the energy budget of the nodes: the node with the highest available energy level in the cluster becomes the head. Such a node is then responsible to transmit the data of all the cluster members to the next tier. The status of such a node remains invariant over time until its energy fully depletes, which leads to selecting a new CH. Although the static approach provides collaboration among the nodes, it puts the energy expenditure burden on the *static* CHs only.

The disaster scenario consists of a varying number of nodes (survivors) randomly distributed over an urban area of 10 by 5 kilometers. Each survivor is equipped with a mobile device (e.g., smartphone) provided with three network interfaces: Bluetooth, WiFi and cellular, with transmission range of 100 m, 200 m, and 500 m correspondingly. By assumption, Bluetooth, WiFi and cellular consume 50 mW, 70 mW and 120 mW respectively [25,26]. Moreover, each node is assigned a random initial energy level in the range of [10 kJ,20 kJ]. Such parameters are summarized in Table 3.

This work focuses on demonstrating how multi-tier architecture can be helpful to relay rescue messages during disasters. Simulations do not focus on the communication channel model and suppose that nodes in the highest tier can communicate with the UAV when they enter in the transmission range of each other. Evaluating the transmission range and the loss probability mainly impacted by the obstacles separating UAV and nodes would be very interesting to investigate in future works. We believe that such an assumption would not impact the overall network communication scheme.

**Table 3.** Summary of used parameters.

| Parameter | Value |
| --- | --- |
| Disaster area | 10 km × 5 km |
| Drone speed | 10 m/s |
| Minimum hovering time | 5 s |
| Drone-$n_3$ tier node data exchange time | 2 s |
| Bluetooth tx range/power consumption | 100 m/50 mW |
| WiFi tx range/power consumption | 200 m/70 mW |
| Cell tx range/power consumption | 500 m/120 mW |

### 6.1.2. Obtained Results

Figure 8 depicts the number of alive nodes over time. More specifically, 400 initial nodes randomly distributed in an urban area disseminate their data in accordance with the three schemes: baseline, static, and the dynamic one. The baseline scheme, in fact, performs poorly in terms of number of alive nodes over time and energy fairness among them: all the nodes die (i.e., no energy availability left) in the network within a relatively short period of time with regard to the golden relief time, and such a trend is almost linear on time. Such is justified by the fact that each node is accountable only for itself, hence it switches on all the necessary network interfaces. This leads to a big number of nodes having their battery depleted at the same time instant. By contrast, the static scheme outperforms the baseline one, leading to a higher number of alive nodes at a given time instant. In fact, the static scheme leads to at least 50 alive nodes more than the baseline scheme and such a gap increases over time. Furthermore, if offers a smoother dynamic of nodes dying in the network, i.e., fewer nodes die at the same time instant. Such a scheme almost doubles the time period within which there is at least one alive node in the network. Our proposed dynamic scheme of clustering and CH selection outperforms the baseline and the static scheme. As it introduces energy expenditure fairness among nodes, it increases the number of alive nodes at a given time instant compared to the other two schemes. Moreover, most of the nodes die alone, or in smaller groups. In fact, such leads to have nodes with comparable energy levels in the network over time. Therefore, such results in the last alive nodes dying together or dying within a short period of time. Such explains the fact that the static scheme slightly outperforms the dynamic one in the last two hours, approximately from hour 6 to 8.

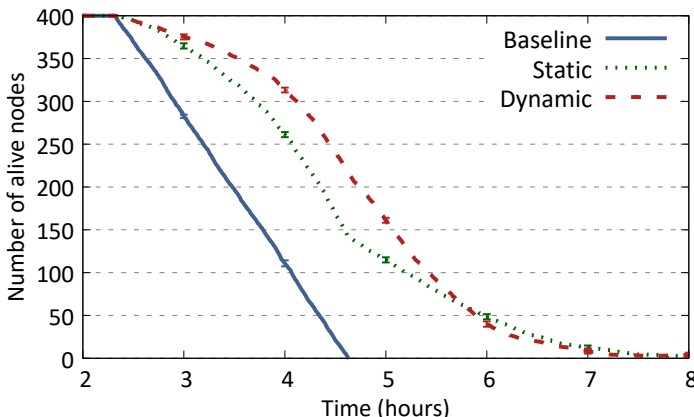

**Figure 8.** Number of alive nodes over time.

Figure 9 depicts the difference between the highest and lowest energy values in the network over time. Similar to Figure 8, the cooperative-based schemes, i.e. static and dynamic, outperform the baseline one, leading to a lower energy difference. That is, the energy levels of the nodes present smaller gaps among each other, i.e., the energy burden put across the nodes is more equally distributed. The initial rise of the difference between the energy levels could be explained by the fact that the nodes with initial low available energy levels deplete their batteries soon after the data dissemination starts.

However, the energy expenditure burden is then equally distributed among the remaining nodes in the dynamic case, hence the difference between the highest and lowest energy values is well below the one of the static case. In fact, the static case puts the energy burden on few nodes that are designated as CHs (whose energy depletes fast), therefore the energy gap between the nodes increases. Moreover, the difference between the energy levels in the baseline case presents a steep slope because all the nodes in the network are responsible to communicate directly with a drone, hence they switch on all the network interfaces at the same time. Though the energy burden is equally distributed, the energy of the nodes depletes almost twice as fast as the cooperative-based schemes.

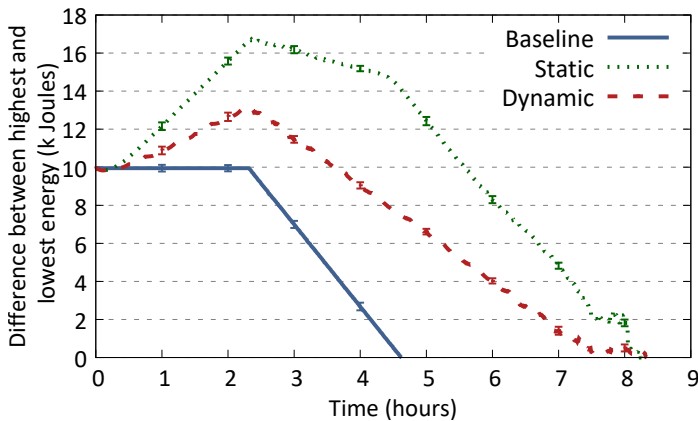

**Figure 9.** Difference between the highest and the lowest energy values in the network.

*6.2. Cooperative Data Relaying with UAVs*

6.2.1. Methodology and Setup

We assess the performance of the proposed TSP and CETSP algorithms with cooperation, namely the TSP-COPE and CETSP-COPE and compare it to the performance of the (*selfish*) TSP and CETSP where no cooperation among nodes holds.

Given a set of points and the corresponding distance between such points as an input, the TSP algorithm generates the shortest path that visits all nodes at least once. Similarly, the CETSP generates the shortest path that visits points that are *close enough* to the original ones. For the sake of clarity, below are summarized all the four path planning algorithms.

- TSP: finds the optimal route that visits each node in the network; there is no cooperation among the nodes to relay data among each other.
- CETSP: determines the minimum number of stops from which the drone can still communicate with all nodes without having to stop at each of them, and further constructs the shortest path that visits all such stops.
- TSP-COPE: similar to TSP, where node cooperation is supported; the optimal route is calculated based on the $n_3$ tier nodes.
- CETSP-COPE: similar to CETSP, where node cooperation is supported; the optimal route is calculated based on the anchor points obtained from the $n_3$ tier nodes.

The following assesses and compares the performance of the two proposed TSP-COPE and CETSP-COPE algorithms (see Section 5) with the selfish TSP and CETSP algorithms. All the four schemes have been implemented as additional modules to the ONE simulator (https://akeranen. github.io/the-one/). The considered scenarios consist of various network densities, where the nodes are randomly situated in an urban area. A drone flies over the disaster area with a speed of 10 m/s. We assume that 2 s is a sufficient time to exchange data with a node, especially given the fact that the data, generally in disaster scenarios, consists of light-weight messages useful for rescue operations and

assistance. However, we consider a *time-guard* of a minimum of 5 s for a drone to hover above a node. Moreover, the hovering time of a drone extends in proportion to the cardinality of the cluster that the node is CH of. For instance, a drone visiting a CH point of a cluster of three members would stop for a time of max (5 s, 6 s). We impose such time-guards to take into account for possible unsuccessful transmissions or collisions.

### 6.2.2. Obtained Results

Figures 10–12 shows the number of stops, the length, and time of a drone tour as a function of the number of nodes in the network for the different scenarios. More specifically, Figure 10 shows how the number of stops of a drone reduces for the two schemes that offer cooperation, i.e., TSP-COPE and CETSP-COPE. In fact, while the number of stops of the TSP scheme with no cooperation increases linearly with the network density, the CETSP-COPE instead shows how the number of stops increases slowly with the network density. Moreover, there is a clear trend, i.e., almost a stable number of stops, for networks with high node density. As such density increases, the multi-tier cooperation among nodes becomes more efficient as nodes have more neighbors, hence more clusters are formed. In fact, the number of stops reduces by more than 70% for a density of 500 nodes. Figures 11 and 12 show how our proposed CETSP-COPE scheme, specifically, outperforms the schemes with no cooperation among the nodes to relay data. This is clearly shown by the fact that the tour length shortens by more than half, and that the drone flying and hovering time reduces by $\approx$ 60% for a high network density. In details, Figure 11 shows that the difference in the tour length for the four schemes increases with the node density; the CETSP-COPE tour length is, in fact, half that of the TSP one. Similarly, the flying and hovering time of a drone shown in Figure 12, increases very slowly for the schemes that provide cooperation, while such values are at least 30% higher for the TSP scheme and high node densities in the network. For instance, the CETSP-COPE drone tour time reduces by around 50% compared to the TSP for 500 nodes in the network. However, CETSP outperforms TSP-COPE and that can be explained by the fact that the cellular transmission range results in a drone to dictate fewer anchor points to visit than the number of $n_3$ tier CHs designated based on node cooperation, hence TSP-COPE.

The energy expenditure of a drone depends mostly on the hovering time [27]. Our cooperative schemes, i.e., TSP-COPE and CETSP-COPE, offer low hovering times compared to flying ones, even for high network densities. Such a trend, presented in Figure 12, shows that the hovering time (proportional to the number of stops) keeps at low levels for all network densities. That is because more clusters with big cardinality are formed as the network density increases, hence a limited number of $n_3$ tier nodes relay the data of all the underlaying tier nodes. Moreover, even the flying (movement) time, which constitutes most of a drone tour time, is further reduced by our cooperative schemes.

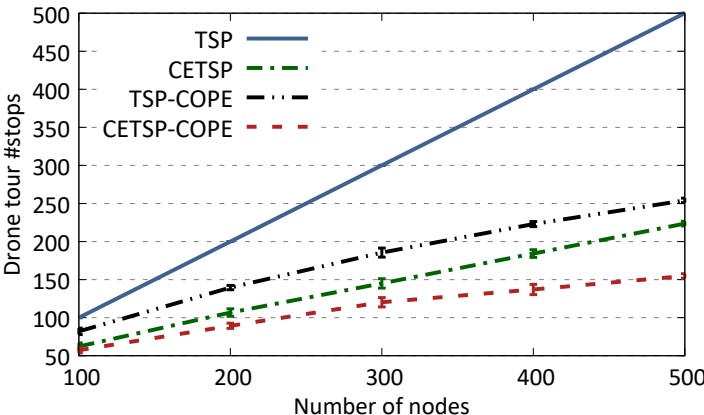

**Figure 10.** Number of stops.

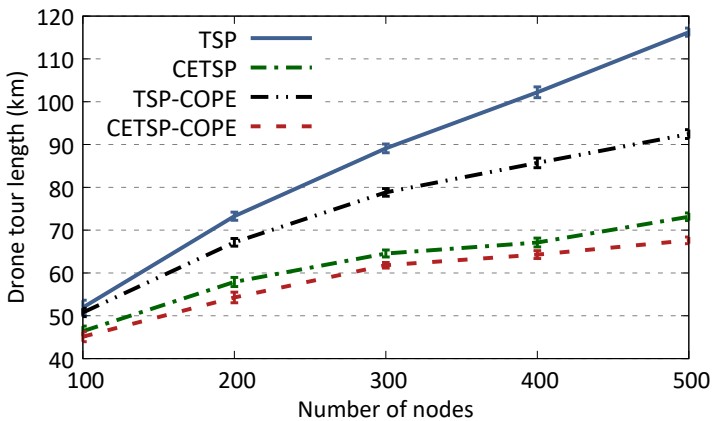

**Figure 11.** Tour length.

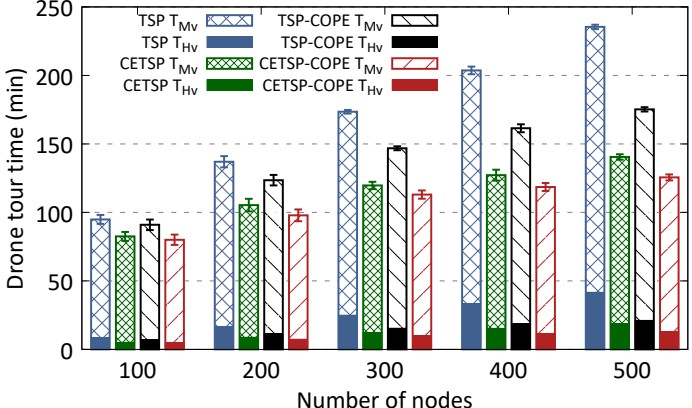

**Figure 12.** Drone hovering and flying time.

## 7. Open Challenges and Future Directions

In this section, we browse a non exhaustive list of some open challenges and potential research directions to investigate to complete and/or improve our architecture efficiency.

### 7.1. Survivor and Rescuer MOBILITY

We have assumed that survivors have a low mobility since they could be wounded and buried under rubble. Our scheme periodically re-computes cliques and time to serve as a representative at each communication layer and thus is assumed to be reliable to faster mobility schemes but this should be better investigated. In addition, as in our approach, all nodes are not all active at the same time, rescuers or drones can be in range of a survivor at a given time but not of its representative during this period and thus messages can be missed.

### 7.2. Belonging to Multiple Cliques

Depending on their connectivity, devices could also be associated with multiple cliques as depicted by Figure 13 with the node $S_2$. However, such an option is out of the scope of this article. Indeed, we simply assume that such nodes will choose to belong only to the smallest clique for balance purpose. Smarter schemes that dynamically adapt the clique membership could be investigated.

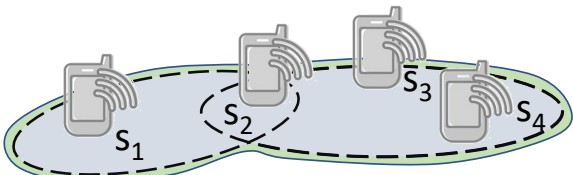

**Figure 13.** Example of clique multi-membership.

### 7.3. Devices Heterogeneity

Our approach is robust to a set of devices featuring different amount of remaining energy. We experimentally verified our assumptions on the possibility to rank communication technologies based on their ranges and costs [28] for a homogeneous set of devices. Nevertheless, as it has been highlighted by a recent study [22], the signal reception quality of a given signal greatly depends of the hardware used and of different settings. Yet, our assumption may not be verified in all cases. But, as it only uses connectivity information between devices to form cliques and so is robust to imperfect propagation ranges and unilateral links. Not verifying this assumption would result in non-optimal energy consumption. So, more investigation should be performed to quantify the impact of this heterogeneity in devices and propagation ranges. In addition, our goal is to rely opportunistically on all available devices that could support the data collection at the rescue center. Therefore, it could include other pieces of infrastructures such as base stations that are still active and powered but disconnected to the core network. They could still act as strong relays since benefiting from an infinite energy reserve. In our scheme, such strong access points will naturally be representative of a clique for all other nodes but our scheme does not leverage its potential longer range and more likely connectivity with several cliques.

### 7.4. Unavailability of Some Communication Interfaces

In our approach, we have assumed that all devices are equipped with all same communication interfaces but for different reasons, this could not be the case. Some interface may be unavailable because the device has not been equipped with it, or it is damaged or the environment does not affect all interfaces similarly. This is thus worth integrating in our scheme the fact that all devices can represent the clique for a given layer.

### 7.5. Multi-Drones

Our scheme currently investigates the use of a single drone and could be simply extended to the use of several drones by sharing between them the areas or anchor nodes to cover. However, due to the dynamics of the anchors at the upmost layers, such a static splitting might not be optimal and a dynamic area responsibility could be set as in [29]. Another interesting technique to be investigated is K-means [34] to jointly derive anchor points and assigning them to each drone.

### 7.6. Dynamic 3D Drones Path Planning

Currently, we assume that thanks to the upmost layer nodes discovery and location, drones are able to compute anchor points and the best traveling path visiting all these anchor points. We also assume that at each visit, nodes inform the drone about change in upmost layer representative allowing it to recompute its path. The path computing is realized in 2 steps: first compute the anchor points and then draw a (Close Enough) traveling salesman problem trajectory. This does not consider the drone autonomy nor a drastic change in representative positions and computes a 2D path. But drone coverage depends on the drone altitude and speed; the drone presents a highly flexible 3-D mobility and the higher the altitude, the larger the coverage but the higher the energy consumption [30]. And this could change the anchor points determination since by flying at a higher altitude, the drone will cover more nodes at a time but will consume more energy. An open problem is thus to determine the best

energy-efficient and minimum-time 3D path to travel the area as fast as possible while still remaining in range of each survivor long enough to assure full servicing. This path should jointly investigate the drone trajectory and the location of the anchor nodes that could dynamically be adapted with drone altitude, while still integrating the pitstop duration at each anchor point, which has a mandatory minimum duration and should be proportional to the number of nodes to serve at this position [31]. A similar work that already provides a good basis with the optimization of several criteria in the trajectory determination is [32,35]. These latter jointly optimize the UAV's flying altitude, antenna beamwidth, UAV's location, and ground terminals' allocated bandwidth. We thus intend to rely on this work to extend it and couple it to our dynamic multi-tier architecture.

### 7.7. Users Devices Recharged by Drones

In our approach, we have considered that the users' devices can only deplete their energy and that the drone only transfer data to them. However, some new approaches suggest that drones can also transfer energy to end devices [33]. The ground multi-tier scheme should dynamically adapt to this characteristic with no modification. However, this could impact the drone behavior since based on the energy it sends to the nodes, it would need to adapt its trajectory.

## 8. Conclusions

This work investigates drone-assisted communications for disaster recovery scenarios. It proposes a dynamic scheme of communication that opportunistically leverages multiple network technologies integrated in to mobile devices, while leveraging the heterogeneity of such devices in terms of available energy levels. Extensive simulations have been conducted and results have shown the benefits of the proposed scheme from both, the drone and the survivors perspective. On the one hand, the proposed scheme allows to maintain a longer and maximum network coverage considering a cooperative scheme which enables, with the support of high-energy nodes, low-energy nodes to preserve their battery for longer time. On the other hand, our proposed solution reduces the energy consumption of the drone by minimizing the number of nodes it visits (i.e. anchor points) and therefore, it reduces the drone path length. However, the advantages of such a solution can be further exploited by introducing a further control parameter – the drone height. Indeed, extending the evaluation presented here to take account of such a parameter is a promising future work.

**Author Contributions:** F.M. and N.M. were responsible for the main idea. They conceived and designed the algorithm, and wrote the manuscript; F.M. carried out the simulations and run the experiments. All authors have read and agreed to the published version of the manuscript.

**Funding:** This work was partly supported by CPER DATA and the Hauts de France PIPA project.

**Conflicts of Interest:** The authors declare no conflict of interest.

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
