# Peer review of "Opportunistic Multi-Technology Cooperative Scheme and UAV Relaying for Network Disaster Recovery"

_information, doi:10.3390/info11010037_

Round 1

Reviewer 1 Report

The paper presents the evaluation of a system for increasing the search and communication with mobile terminals during disaster scenarios. The system is based on the use of an application for cooperative communications that exploits WiFi and BT for relaying to a UAV for external rescue and connectivity.

The work is well organized. I think that authors should explain better the methodology followed for obtaining the numerical results, which seems partially experimental (for the transmission range) and simulative. In particular, it would be important to know if, in the simulations:

what channel model has been used; if there is a loss probability of the packet between terminal and UAV.

Author Response

The paper presents the evaluation of a system for increasing the search and communication with mobile terminals during disaster scenarios. The system is based on the use of an application for cooperative communications that exploits WiFi and BT for relaying to a UAV for external rescue and connectivity.

The work is well organized. I think that authors should explain better the methodology followed for obtaining the numerical results, which seems partially experimental (for the transmission range) and simulative. In particular, it would be important to know if, in the simulations: what channel model has been used; if there is a loss probability of the packet between terminal and UAV.

---

We would like to thank the reviewer for his/her valuable comments.

Conducted simulations focused on demonstrating how multi-tier architecture can be helpful to relay rescue messages during disaster. We did not focus on the communication channel model and we supposed that nodes can communicate with the UAV when they enter in the transmission range of each other. However, we have highlighted its importance on the loss probability and added some details in Section 6.
We believe that such an assumption would not impact the overall network communication scheme.

Reviewer 2 Report

This paper investigates the UAV relaying for network disaster recovery. The topic is timely and interesting. The reviewer may have the following concerns:
1. Since only one drone is considered in the system model, the discussion about multiple UAVs is needed. One way to do that is to divide the network into multiple subareas by using the K-means as in Energy Efficient Resource Allocation in UAV-Enabled Mobile Edge Computing Networks.
2. In wireless communications, the mobility of the users is not considered in the network architecture.
3. The analysis of this paper is lacked. It would be helpful if the complexity analysis of Algorithm 1 can be provided.
4. In the related work, only the converge problem in UAV is investigated. However, due to the mobility of UAV, it is also of importance to investigate the capacity performance/multiple access issue as in [1-4].
[1] Joint Trajectory and Communication Design for Secure UAV Networks
[2] Energy Efficient UAV Communication with Energy Harvesting
[3] Energy Efficient Resource Allocation in UAV-Enabled Mobile Edge Computing Networks
[4] Joint altitude, beamwidth, location, and bandwidth optimization for UAV-enabled communications
5. In the simulations, the effect of drone height is not shown.
6. The language of this paper can be improved. For example, in the abstract, "A multi-tiers network architecture" should be "A multi-tier network architecture".

Author Response

We would like to thank the reviewer for his/her valuable comments. We believe they helped us to greatly improve our paper. 

--------------
Q1. Since only one drone is considered in the system model, the discussion about multiple UAVs is needed. One way to do that is to divide the network into multiple subareas by using the K-means as in Energy Efficient Resource Allocation in UAV-Enabled Mobile Edge Computing Networks.
R1. Many thanks for this comment and for providing potential research direction for multi-drone case. Indeed, in this first approach, we have assumed the use of a single drone but the extension to multiple drones is an interesting one that we have discussed in Section 7.5. In this revised version, we have enriched this section with the reviewer's suggestion.
--------------
Q2. In wireless communications, the mobility of the users is not considered in the network architecture.
R2. Indeed, in the evaluation of our architecture, we have assumed a very low mobility from the users, that can be motivated by the scenario we considered (users may be injured or/and blocked so they can not move fast). But we agree our system should be evaluated under different mobility models, which we detail in discussion part, section 7.1.
--------------
Q3. The analysis of this paper is lacked. It would be helpful if the complexity analysis of Algorithm 1 can be provided.
R3. Thanks for your pertinent comment. Actually, the complexity is rather low. Indeed, a node only has to identify cliques within its neighborhood (in $O(n.log(n))$ if $n$ is the number of neighbors) and sort the nodes of the clique according to their Id, so in $O(m.log(m))$ where $m$ is the number of nodes in the clique. Determining close bounds for $n$ and $m$ is a very interesting study that we may complete finely. Nevertheless, an upper bound for $m$ is $n$ ($m\leq n$). So an upper bound for the complexity is $O(n.log(n))$. We have added a note about it in the paper.
--------------
Q4. In the related work, only the converge problem in UAV is investigated. However, due to the mobility of UAV, it is also of importance to investigate the capacity performance/multiple access issue as in [1-4].
[1] Joint Trajectory and Communication Design for Secure UAV Networks
[2] Energy Efficient UAV Communication with Energy Harvesting
[3] Energy Efficient Resource Allocation in UAV-Enabled Mobile Edge Computing Networks
[4] Joint altitude, beamwidth, location, and bandwidth optimization for UAV-enabled communications

R4. Thanks for pointing this out. We have added these references in the state of the art section and where it is relevant.
--------------
Q5. In the simulations, the effect of drone height is not shown.
R5. Thanks for your comment. Indeed, our simulations have not considered so far the height effect. In this paper, we draw a first architecture that is evaluated assuming different simplifying assumptions such as the drone's height. However, we are aware of this point and discuss it in Section 7.6 that we have completed for highlighting this point.
--------------
Q6. The language of this paper can be improved. For example, in the abstract, "A multi-tiers network architecture" should be "A multi-tier network architecture".
R6. Thanks for your comment. We have proof-read the document.
--------------

Round 2

Reviewer 2 Report

Thank you for the response. The reviewer's concerns has been well solved.